# Atmospheric impregnation behavior of calcium phosphate materials for antibiotic therapy in neurotrauma surgery

**Akihito Kato** *

Department of Emergency, Disaster, and Critical Care Medicine, Showa University of Medicine, Tokyo, Japan

* akihito@med.showa-u.ac.jp

## Abstract

As part of a verification model of antibiotic therapy in cranioplasty, we evaluated the impregnation efficiency of interporous calcium phosphate materials with saline under atmospheric pressure and compared it to the efficiency of using the decompression method established by the Japanese Industrial Standard, under which pressure is reduced by 10 kPa. Five types of material formed in 1 mL cubes were selected as test samples: two consisting of hydroxyapatite (HAp) with 85% and 55% porosity and three of β-tricalcium phosphate (β-TCP) with 75%, 67%, and 57% porosity. All test samples showed an impregnation ratio of more than 70%, except for the HAp sample with 55% porosity, which had a ratio of approximately 50%. These high ratios were achieved at only 15 min. The impregnation effects were likely dependent on porosity and were independent of base material, either HAp or β-TCP. Obtaining sufficient impregnation and antimicrobial efficacy in materials with low porosity, which are commonly used in cranioplasty, would require an increased volume of antibiotics rather than increased duration of impregnation. Our findings will enable the simple preparation of drug-impregnated calcium phosphate materials, even in operating rooms not equipped with a large decompression device.

**Data Availability Statement:** All relevant data are within the manuscript and its Supporting Information files.

**Funding:** The authors have no funding to report.

## Introduction

The prognosis of patients treated for traumatic brain injury is strongly dependent on prehospital care, maintenance of emergency systems, and rapid response to secondary injuries such as blood flow and metabolic abnormalities [1,2]. In particular, open depressed cranial fractures require surgery within 24 h after onset, and the risk of infection clearly increases after 48 h [3]. While the incidence of postoperative infection is approximately 5% of cases in the orthopaedic field, it is 11–34% in the neurosurgical field [4–6]. This difference is considered due to the typically more complicated and larger craniotomy required for the latter cases, and the greater contact between bone fragments and the brain. Local drug delivery devices provide more efficient delivery of larger amounts of antibiotic to sites of infection [7]. While most orthopaedic implants are performed using antibiotic-impregnated bone cement [8], the mixing and shaping of bone cement during cranioplasty is difficult when the bone defect is large or cosmetic

**Competing interests:** The authors have declared that no competing interests exist.

issues are present. For this reason, custom-made artificial bone is often used, composed of interporous calcium phosphate such as hydroxyapatite (HAp) and β-tricalcium phosphate (β-TCP). Antibiotics are generally impregnated into the pores using a centrifuge or vacuum. Ito-kazu et al. impregnated the antibiotic arbekacin into an HAp block by centrifugation at 1500 rpm for 15 min, and demonstrated that the minimum inhibitory concentration (MIC) for methicillin-resistant *Staphylococcus aureus* was maintained unchanged for 18 days [9]. How-ever, this method is difficult to perform in the operating room and limits the size of the HAp block. Furthermore, these authors also impregnated the antibiotic isepamicin (ISP) into an HAp block using a vacuum method for 20 min, and demonstrated that the MIC against the common causative organisms of osteomyelitis was maintained unchanged for 18 days [10,11]. However, these devices are not available in all operating rooms. Further, if antibiotic impreg-nation is attempted under atmospheric pressure, the correlation between duration and level of antibiotic impregnation is unclear.

Here, we evaluated the behavior of calcium phosphate materials impregnated under atmo-spheric conditions without special equipment, as shown in Fig 1, and compared results with those obtained using the standardized decompression method established by the Japanese Industrial Standard (JIS).

## Materials and methods

### Materials

Calcium phosphate materials were obtained from HOYA Technosurgical (Tokyo, Japan) and are standardized for clinical use [12–14]. Five types were selected as test samples, namely CP-A, CP-B, CP-C, CP-D, and CP-E. CP-A (APACERAM®-AX) and CP-E (APACERAM®) consist of pure HAp with 85% and 55% porosity, respectively, while CP-B (SUPERPORE® standard type), CP-C (SUPERPORE® hard type) and CP-D (SUPERPORE® EX) consist of pure β-TCP with 75%, 67%, and 57% porosity, respectively. All test samples are formed in 1 mL cubes (10 mm × 10 mm × 10 mm), except for CP-C, which is formed in a 1.5 mL cuboid (10 mm × 10 mm × 15 mm).

### Decompression method

Impregnation under decompression was performed according to the JIS A1509-3 standards using the impregnation system shown in Fig 2. Each test sample was placed within a vessel in a vacuum state for 30 min during which the pressure was reduced by 10 kPa. Subsequently, saline was injected into the vessel to a depth of 50 mm, while the vacuum state was kept intact. The vacuum was then removed and the vessel was left to stand under atmospheric pressure for 15 min. The test samples were then wiped gently with a wet towel and weighed. Impregnation amount per gram of test sample ($W_r$) was then calculated using the following formula:

$$W_r = \frac{W_a - W_b}{W_b} \tag{1}$$

where $W_a$ and $W_b$ indicate the test sample weights after and before impregnation, respectively.

### Atmospheric method and impregnation ratio

Impregnation under atmospheric pressure was carried out in the same manner as above except that the samples were not placed in a vacuum and the pressure was not reduced. The test sam-ples were removed and weighed at 15, 30, 60, and 120 min after saline injection. Impregnation amount per gram under atmospheric pressure ($W_s$) was calculated as above, and the

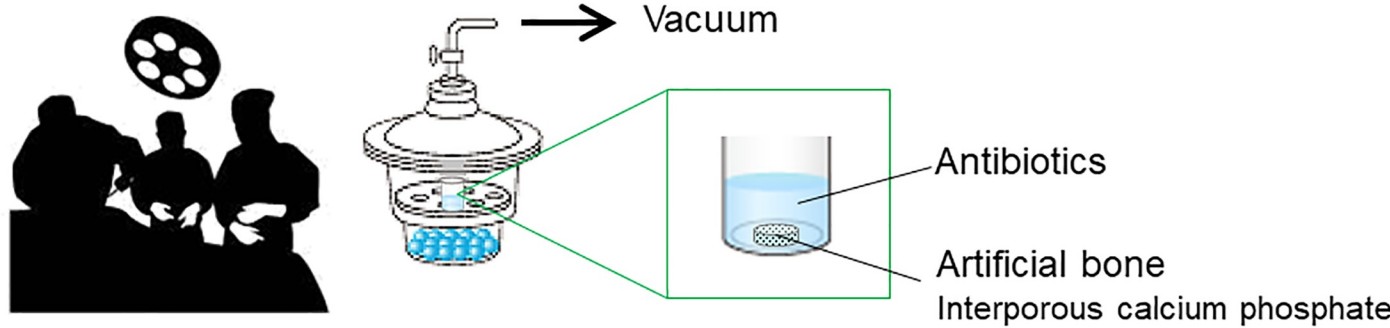

# Antibiotics impregnation methods for implants to prevent postoperative infection

Vacuum

Antibiotics

Artificial bone
Interporous calcium phosphate

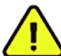 These devices are not available in all operating room.

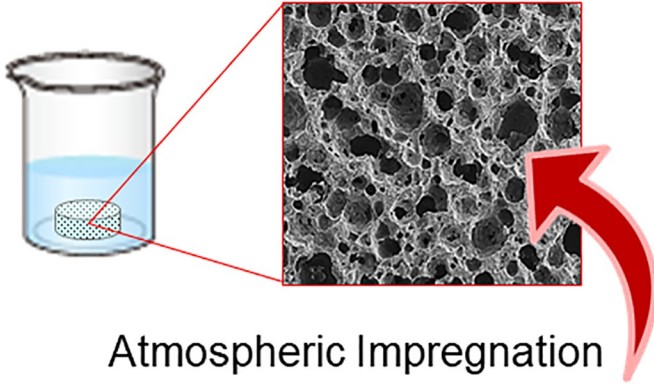

Impregnation ratio: more than 70%
(compared to the decompression method)

Duration of impregnation: only 15 min

Atmospheric Impregnation

**Fig 1. Schematic illustration of this study.** Antibiotics are generally impregnated into the pores of implants using a vacuum to prevent postoperative infection. However, such decompression devices are not available in all operating rooms. We demonstrated that an impregnation ratio of more than 70% was achieved under atmospheric conditions at only 15 min.

impregnation ratio ($R_s$) was calculated as follows:

$$R_s = \frac{W_s}{W_r} \times 100 \tag{2}$$

The experimental impregnation ratio per volume ($R_e$) of test sample under reduced pressure and atmospheric pressure, where $v$ and $d$ are the volume of each test sample and the specific gravity of saline, respectively, was calculated as follows:

$$R_e = \frac{(W_a - W_b)}{vd} \times 100 \tag{3}$$

The actual porosity ($R_t$) of each sample, which corresponds to the theoretical impregnation

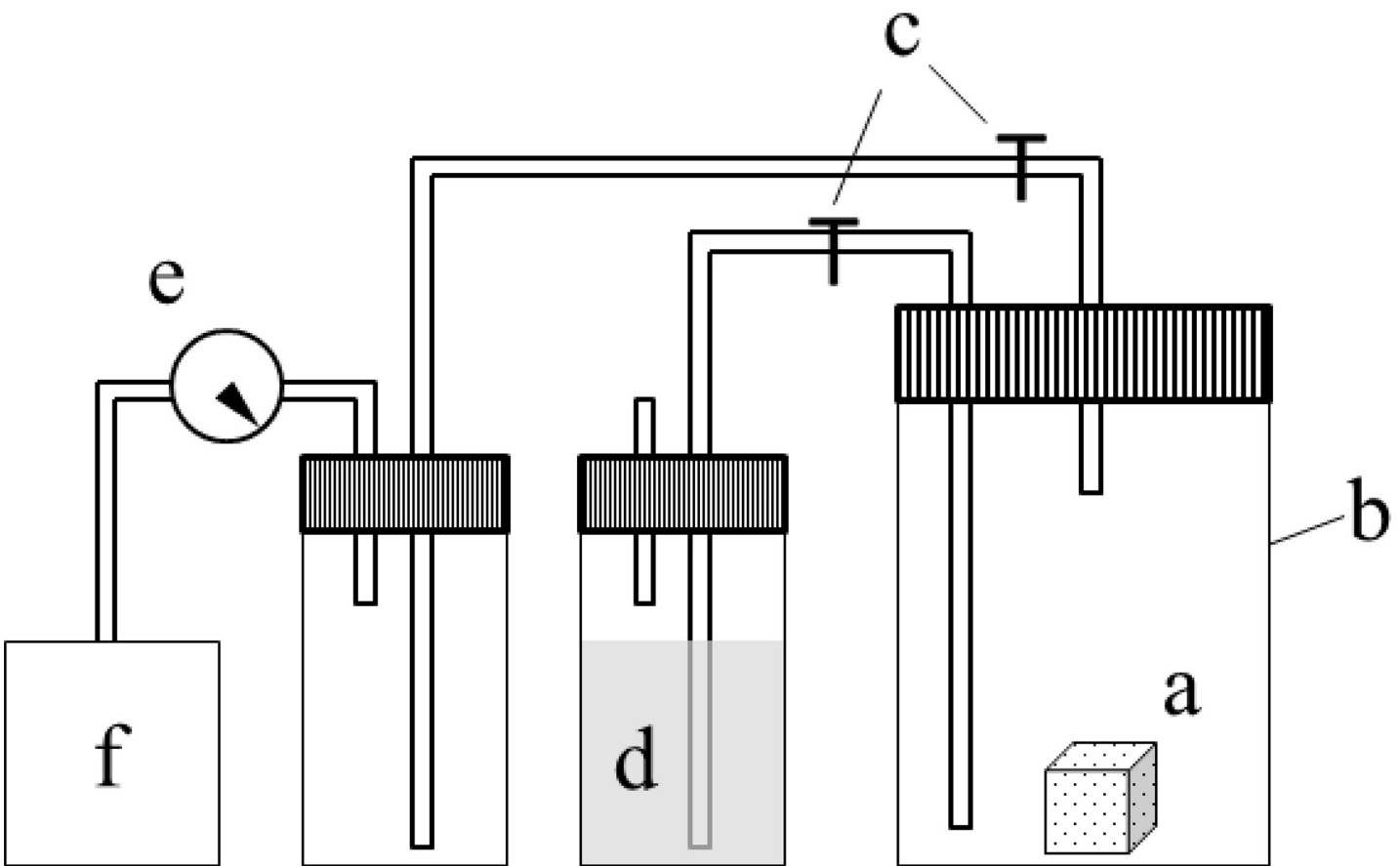

**Fig 2. The impregnation test system under decompression.** (a) Calcium phosphate test specimen; (b) pressure-resistance glass vessel; (c) cock; (d) saline; (e) pressure gauge; and (f) vacuum pump.

ratio, was calculated using following formula:

$$R_t = \left(1 - \frac{W}{vd}\right) \times 100 \tag{4}$$

where $W$, $v$, and $d$ are the weight, volume, and specific gravity of each test sample, respectively. All experiments were replicated three times at room temperature, and mean values and standard deviations are shown.

## Statistics

Impregnation ratios at set times were analyzed with the Mann-Whitney U-test as a non-parametric test using HAD software (version 16, Kwansei Gakuin University, Hyogo, Japan) [15]. This test was used because the normality was unreliable due to the small sample size, and comparison between two groups, such as 15 min versus other durations (duration of impregnation) or reduced versus atmospheric pressure (impregnation method), was sufficient to evaluate impregnation efficiency. Differences were considered statistically significant when $p$-values were less than 0.05.

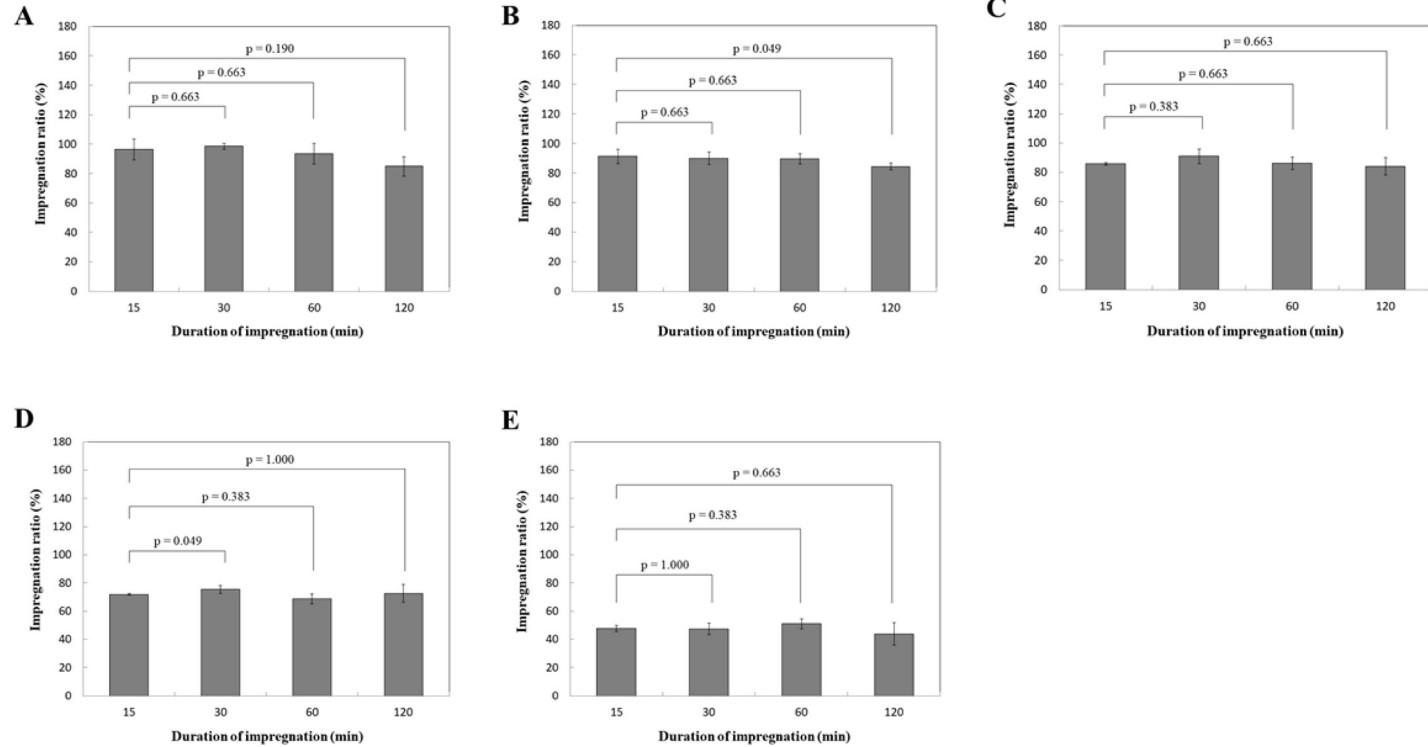

**Fig 3. Impregnation ratio of five types of calcium phosphate material under atmospheric pressure compared to the decompression method standardized by JIS.**
(A) CP-A consisting of HAp with 85% porosity; (B) CP-B consisting of β-TCP with 75% porosity; (C) CP-C consisting of β-TCP with 67% porosity; (D) CP-D consisting of β-TCP with 57% porosity; and (E) CP-E consisting of HAp with 55% porosity.

## Results

Fig 3 shows the impregnation ratio under atmospheric pressure ($R_s$). All of the test samples showed an impregnation ratio of more than 70%, except for CP-E, which had a ratio of approximately 50%. These high ratios were achieved at only 15 min, after which they plateaued, except for CP-B and CP-D, which showed a slight decrease between 15 and 120 min and a slight increase between 15 and 30 min, respectively.

Fig 4 shows the experimental impregnation ratio at 15 min per volume of test sample under reduced pressure and atmospheric pressure ($R_e$), and the theoretical impregnation ratio ($R_t$) if all the pores of each test sample were filled with saline. The ratio in the reduced group was significantly higher than that in the atmospheric group ($p = 0.049$ for all test samples), and the impregnation ratio did not reach the theoretical value, even when impregnated under reduced pressure. CP-E showed a markedly lower impregnation ratio than the other samples.

## Discussion

The impregnation of calcium phosphate materials of various porosities with saline under atmospheric pressure was evaluated and the results were compared to those using the JIS decompression method. Given that the results of this study depend on the structure and physical properties of the test sample, the findings will only be applicable to the materials used in this study. Because these materials are standardized for porosity, density, purity, and mechanical strength as medical devices [12–14], the results will be reproducible within these limits. The test samples all showed similar impregnation ratios at 15 min, except for low porosity

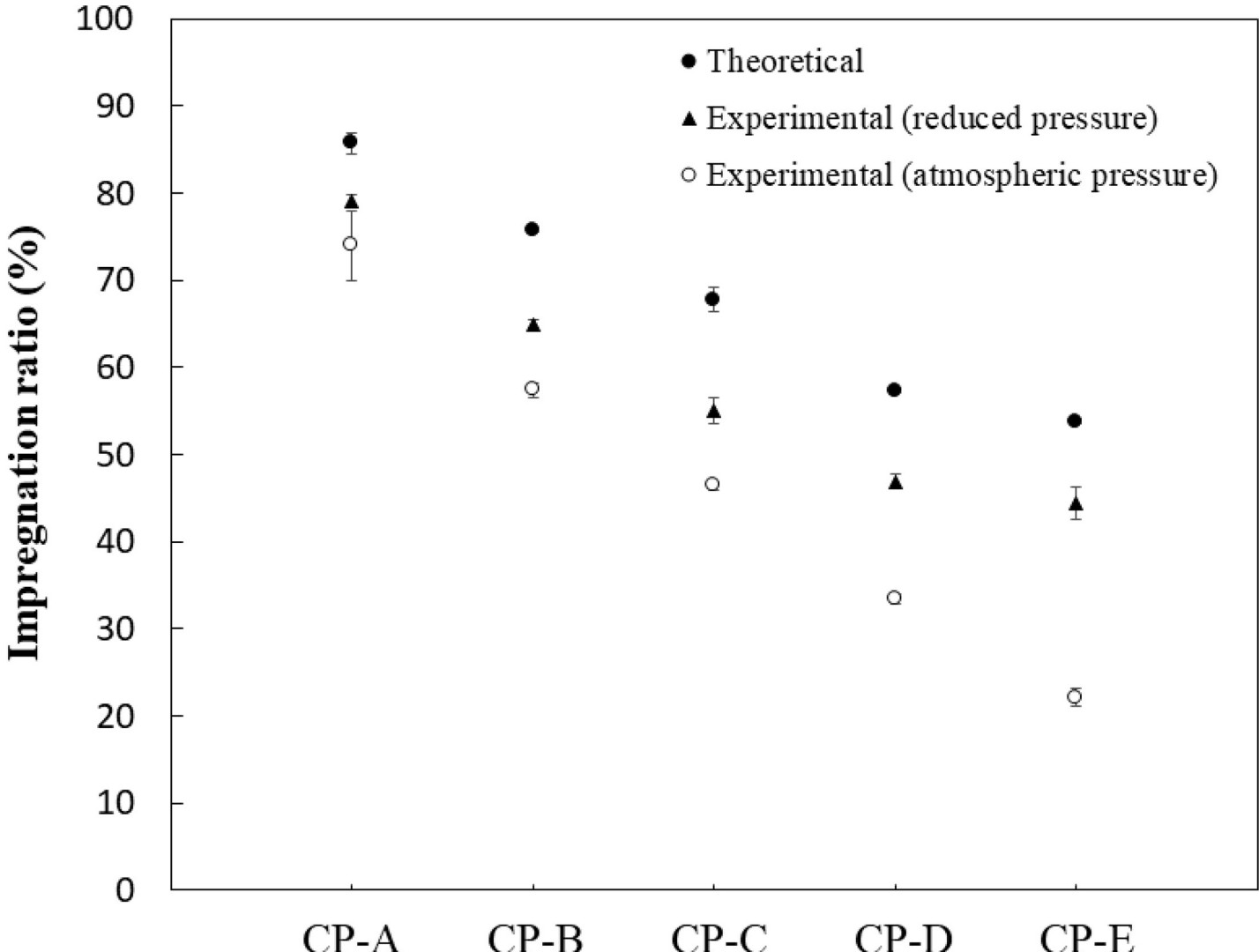

**Fig 4. Comparison of theoretical and experimental impregnation ratios at 15 min per volume of test sample under reduced pressure and atmospheric pressure.**
While all data are presented with error bars, some of the error bars are difficult to see because the magnitude of the error is smaller than the marker size (e.g. CP-E in the theoretical group).

CP-E (Fig 3). There were no significant differences in impregnation level among the materials, even when the test was continued for longer than 15 min, except for CP-B between 15 and 120 min and CP-D between 15 and 30 min. CP-B consists of β-TCP with high porosity, which may cause slight dissolution, resulting in a reduced weight. CP-D would require a greater duration of impregnation for a plateau to occur, because it has interconnecting pores despite its low porosity [12], resulting in an increased weight.

The impregnation ratio per volume of each test sample should theoretically be the same as the porosity of the sample. However, we found a significant difference between the experimental and theoretical impregnation ratios, indicating that not all of the pores of a sample were filled with saline even when impregnated under a reduced pressure (Fig 4). These results show that the interporous calcium phosphate material has closed pores which the external fluid cannot enter [16]. Figs 3 and 4 show that CP-E has a markedly lower impregnation ratio than the other samples. One explanation for this may be that CP-E has fewer interconnecting pores

within the material compared with CP-A, CP-B, CP-C, and CP-D, which have a triple pore structure (macro pores, micro pores, and interconnecting pores) [12]. The difference in impregnation ratio between CP-D and CP-E despite their similar porosity may be due to differences in such microstructures within the materials. Generally, drug impregnation into calcium phosphate materials is carried out by centrifugation and vacuum methods, which forcibly removes air inside the material to increase the impregnation effect [9–11]. To obtain an equivalent effect without decompression, removal of air from inside the material would represent a rate-limiting step of the process. Air bubbles inside the material would be removed by exchange with external fluid, for which the following Stokes' equation could be applied [17]:

$$v = \frac{D^2(\rho_b - \rho_f)g}{18\eta} \tag{5}$$

where $v$ and $D$ are the velocity and size of bubble, respectively, $\rho_b$ and $\rho_f$ are the density of bubble and fluid, respectively, $g$ is the gravity acceleration, and $\eta$ is the viscosity. The removal efficiency of air is proportional to the square of the pore size, assuming that the bubble and pore sizes are equal, and the impregnation effects depend on both porosity and interconnecting pores and are independent of base material, HAp or β-TCP.

Itokazu et al. demonstrated that when the antibiotic ISP was impregnated into an HAp block with 50% porosity using a vacuum method for 20 min and the pressure was reduced by approximately 250 mmHg (33 kPa), the resulting ISP-impregnated HAp block with a 40% impregnation ratio released ISP for more than 18 days [10]. If the elution curve is integrated from day 2 to 18, however, the total elution volume of ISP appears to exceed the impregnated amount before the examination. We consider that it is unclear whether the antibiotic fraction impregnated using decompression can be released naturally from interporous materials, and that it is necessary to consider the effects of the fraction adsorbed to the material surface [18].

In conclusion, we demonstrated that atmospheric impregnation of saline into calcium phosphate materials, except in a test sample with fewer interconnecting pores, achieved an impregnation ratio of more than 70% at 15 min using a standardized decompression method established by the JIS. The impregnation effects likely depend on both the porosity and interconnecting pores. Obtaining sufficient impregnation and antimicrobial efficacy in materials with low porosity or fewer interconnecting pores, which are commonly used in cranioplasty, would require an increased volume of antibiotics rather than increased duration of impregnation. In our neurotrauma surgery, antibiotic-impregnated calcium phosphate materials are prepared within approximately one hour from the onset to completion of the craniotomy. Accordingly, the completion of impregnation within 15 min, as shown in this study, would have no effect on neurotrauma surgery, and the sintered HAp material shows little dissolution to affect physical properties and mechanical strength in such a short duration of impregnation [19]. Our findings will enable the simple preparation of drug-impregnated calcium phosphate materials, and could also be applied to the preservation of bone fragments impregnated with drug solution in craniotomy, even in operating rooms without a large decompression device.

## Supporting information

**S1 File. Raw data.**
(PDF)

## Author Contributions

**Conceptualization:** Akihito Kato.

**Formal analysis:** Akihito Kato.

**Funding acquisition:** Akihito Kato.

**Investigation:** Akihito Kato.

**Methodology:** Akihito Kato.

**Project administration:** Akihito Kato.

**Resources:** Akihito Kato.

**Supervision:** Akihito Kato.

**Validation:** Akihito Kato.

**Visualization:** Akihito Kato.

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
