## [Decision Letter · Decision Letter 0]

13 Jan 2020

PONE-D-19-30682

Atmospheric impregnation behavior of calcium phosphate materials for antibiotic therapy in neurotrauma surgery

PLOS ONE

Dear Dr. KATO,

Thank you for submitting your manuscript to PLOS ONE. After careful consideration, we feel that it has merit but does not fully meet PLOS ONE’s publication criteria as it currently stands. Therefore, we invite you to submit a revised version of the manuscript that addresses the points raised during the review process.

Specifically:

-Incomplete review of the previous research

-Incomplete description of experimental procedure

-Incomplete statistical analysis of the experimental data

-Limited experimental data in support of the hypothesis

We would appreciate receiving your revised manuscript by Feb 27 2020 11:59PM. To enhance the reproducibility of your results, we recommend that if applicable you deposit your laboratory protocols in protocols.io, where a protocol can be assigned its own identifier (DOI) such that it can be cited independently in the future. For instructions see: http://journals.plos.org/plosone/s/submission-guidelines#loc-laboratory-protocols

We look forward to receiving your revised manuscript.

Kind regards,

Esmaiel Jabbari, PhD

Academic Editor

PLOS ONE

2. We note you have not completed the financial disclosure section.

a)    Please provide a Funding Statement that declares *all* the funding or sources of support received during this specific study (whether external or internal to your organization) as detailed online in our guide for authors at http://journals.plos.org/plosone/s/submit-now.  

b)    Please state what role the funders took in the study.  If any authors received a salary from any of your funders, please state which authors and which funder. If the funders had no role, please state: "The funders had no role in study design, data collection and analysis, decision to publish, or preparation of the manuscript."

3. We note you have not completed the Competing Interests section:  

Reviewers' comments:

Reviewer's Responses to Questions

**Comments to the Author**

1. Is the manuscript technically sound, and do the data support the conclusions?

Reviewer #1: Partly

Reviewer #2: Yes

2. Has the statistical analysis been performed appropriately and rigorously? 

Reviewer #1: No

Reviewer #2: Yes

3. Have the authors made all data underlying the findings in their manuscript fully available?

Reviewer #1: Yes

Reviewer #2: Yes

4. Is the manuscript presented in an intelligible fashion and written in standard English?

Reviewer #1: Yes

Reviewer #2: Yes

5. Review Comments to the Author

Reviewer #1: (1) In the Introduction Section, a brief review of the literature is lacking. Has any work been reported on this topic? If so, what are the shortcomings of these studies?

(2) Throughout the manuscript, it best to replace "time" with "duration of impregnation".

(3) In the Materials and Methods Section, it was stated that tests were conducted using (a) special equipment, under atmospheric impregnation and (b) without special equipment, under atmospheric condition. However, it appears that the only results given as those obtained using special equipment, under atmospheric impregnation.

(4) Why was a parametric method (unpaired t-test) used for the statistical analysis? It should be noted that before using a parametric method, normality must be established for each of the datasets. Thus, it is best to use a non-parametric method, such as the Kruskal-Wallis method with an appropriate post-hoc test (such as Bonferroni).

(5) With regard to Figure 4, there is some confusion with regard to error bars:

(a) why are there error bars on the theoretical results?

(b) why are error bars not shown in some of the experimental results?

(6) In the Results Section, why are there no results of the statistical comparison between impregnation ratios under atmospheric pressure [ATMOS RATIO) versus when the decompression machine was used (per JIS A1509-3 Standard) [DECOMP RATIO]?

(7) As a follow-up to item (6), from the results presented in Figure 4, it appears that, by and large,

for a given CP, DECOMP RATIO is markedly higher than ATMOS RATIO. Thus, the results do not support the authors' assertion/hypothesis.

(8) In the Discussion Section, please add statements about study limitations. For each limitation, state (a) why it was used; and (b) why its use does not undermine the conclusion reached in the study.

(9) As a general comment, the work presented in this manuscript is very limited in its scope. For example, how does duration of impregnation affect physical, mechanical, and other properties of the CaP-based composite that are crucial for use in neurotrauma surgery (ATMOS group versus DECOMP group)? What is the impact on duration of the surgery?

This shortcoming of the work reported in this manuscript makes it very difficult to assess the extent to which the present findings will influence surgical practice (if at all).

Reviewer #2: This paper is acceptable after the minor revision.

Difference in impregnation ratio between hydroxyapatite and b-TCP with similar porosity should be discussed in more detail, especially D and E in Fig. 3.

6. PLOS authors have the option to publish the peer review history of their article (what does this mean?). If published, this will include your full peer review and any attached files.

Reviewer #1: No

Reviewer #2: No

---

## [Author Response · Author response to Decision Letter 0]

27 Feb 2020

RESPONSE TO EDITOR:

We wish to express our appreciation to the Editor for his or her insightful comments, which have helped us significantly improve the paper.

Comment 1: Please ensure that your manuscript meets PLOS ONE's style requirements, including those for file naming.

Response: We appreciate the editor’s comment. We have confirmed that our manuscript meets PLOS ONE’s style requirements. 

Comments 2: We note you have not completed the financial disclosure section. (snip) Please include your amended statements within your cover letter; we will change the online submission form on your behalf.

Response: We appreciate the editor’s comment. We have added the financial statement within our cover letter. 

Comment 3: We note you have not completed the Competing Interests section. Please complete your Competing Interests on the online submission form to state any Competing Interests. (snip) This information should be included in your cover letter; we will change the online submission form on your behalf.

Response: We appreciate the editor’s comment. We have added the Competing Interests statement within our cover letter and on the online submission form.

Comment 4: PLOS requires an ORCID iD for the corresponding author in Editorial Manager on papers submitted after December 6th, 2016. Please ensure that you have an ORCID iD and that it is validated in Editorial Manager.

Response: In accordance with the editor’s comment, we have created a new ID.

Thank you again for your comments on our paper. We trust that the revised manuscript is suitable for publication.

RESPONSE TO REVIEWER 1:

We wish to express our appreciation to the Reviewer for his or her insightful comments, which have helped us significantly improve the paper.

Comment 1: In the Introduction Section, a brief review of the literature is lacking. Has any work been reported on this topic? If so, what are the shortcomings of these studies?

Response: We appreciate the reviewer’s comment. We agree that this point requires clarification, and have added the following text with some references to the Introduction section (p. 3, lines 53-60):

“Itokazu et al. impregnated the antibiotic arbekacin into an HAp block by centrifugation at 1500 rpm for 15 min, and demonstrated that the minimum inhibitory concentration (MIC) for methicillin-resistant Staphylococcus aureus was maintained unchanged for 18 days [9]. However, this method is difficult to perform in the operating room and limits the size of the HAp block. Furthermore, these authors also impregnated the antibiotic isepamicin (ISP) into an HAp block using a vacuum method for 20 min, and demonstrated that the MIC against the common causative organisms of osteomyelitis was maintained unchanged for 18 days [10,11].”

Comment 2: Throughout the manuscript, it best to replace "time" with "duration of impregnation".

Response: In accordance with the reviewer’s comment, we have replaced “time” with “duration of impregnation” throughout the manuscript.

Comment 3: In the Materials and Methods Section, it was stated that tests were conducted using (a) special equipment, under atmospheric impregnation and (b) without special equipment, under atmospheric condition. However, it appears that the only results given as those obtained using special equipment, under atmospheric impregnation.

Response: We appreciate the reviewer’s comment. But, we believe the reviewer is mistaken on this admittedly difficult point. This study compares the impregnation methods under atmospheric and reduced pressure (p. 4, lines 63-66). Special equipment was used to create decompression conditions, however, the same equipment was used for experiments under atmospheric pressure, since it was necessary to experiment under the same conditions except for pressure. Therefore, in the Materials and Methods section, we intend to state that tests were conducted using (a) special equipment, under atmospheric pressure and (b) special equipment, under reduced pressure, and we believe that the results were correctly described in the Results section. As shown in Figure 2, this special equipment is just a vessel without a decompression operation, therefore, we considered that the results under condition (a) above represent those in an operating room without special decompression devices, and wish to retain the original text.

Comment 4: Why was a parametric method (unpaired t-test) used for the statistical analysis? It should be noted that before using a parametric method, normality must be established for each of the datasets. Thus, it is best to use a non-parametric method, such as the Kruskal-Wallis method with an appropriate post-hoc test (such as Bonferroni).

Response: The reviewer’s comment is correct. Thank you for introducing the recommended method. We agree that this point requires clarification, and need to retry the statistical analysis. Therefore, we have added the following text to the Statistics section (p. 7, lines 131-136):

“Impregnation ratios at set times were analyzed with the Mann-Whitney U-test as a non-parametric test using HAD software (version 16, Kwansei Gakuin University, Hyogo, Japan) [15]. This test was used because the normality was unreliable due to the small sample size, and comparison between two groups, such as 15 min versus other durations (duration of impregnation) or reduced versus atmospheric pressure (impregnation method), was sufficient to evaluate impregnation efficiency.”

Comment 5: With regard to Figure 4, there is some confusion with regard to error bars:

(a) why are there error bars on the theoretical results?

(b) why are error bars not shown in some of the experimental results?

Response: We appreciate the reviewer’s comment. (a) As described in the Materials and Methods section (p. 6, lines 120-121), the actual porosity of each sample corresponds to the theoretical impregnation ratio. Therefore, in this study, the porosity of each sample was actually measured using the value of the weight, volume, and specific gravity, and was calculated with equation (4). (b) In Figure 4, there were some datasets where the width of the error bar was smaller than that of the marker circle, which showed a result like no error bar. Accordingly, we tried to modify Figure 4 with smaller marker sizes. However, further reduction in marker size decreases the visibility of the data, therefore, we have added the following text to the Figure caption (p. 8, lines 160-162):

“While all data are presented error bars, some of the error bars are difficult to see because the magnitude of the error is smaller than the marker size (e.g. CP-E in the theoretical group).”

Comment 6: In the Results Section, why are there no results of the statistical comparison between impregnation ratios under atmospheric pressure [ATMOS RATIO] versus when the decompression machine was used (per JIS A1509-3 Standard) [DECOMP RATIO]?

Response: We appreciate the reviewer’s comment. We agree that this point requires clarification, and need to try the statistical analysis. Therefore, we have added the following text to the Results section (p. 8, lines 148-149):

“The ratio in the reduced group was significantly higher than that in the atmospheric group (p = 0.049 for all test samples),”

Comment 7: As a follow-up to item (6), from the results presented in Figure 4, it appears that, by and large, for a given CP, DECOMP RATIO is markedly higher than ATMOS RATIO. Thus, the results do not support the authors' assertion/hypothesis.

Response: The reviewer’s comment is correct. To clarify, we have changed the description to the following text in the Discussion section (p. 11, lines 211-214):

“In conclusion, we demonstrated that atmospheric impregnation of saline into calcium phosphate materials, except in a test sample with fewer interconnecting pores, achieved an impregnation ratio of more than 70% at 15 min using a standardized decompression method established by the JIS.”

Comment 8: In the Discussion Section, please add statements about study limitations. For each limitation, state (a) why it was used; and (b) why its use does not undermine the conclusion reached in the study.

Response: We appreciate the reviewer’s comment. Accordingly, we have added the following text to the Discussion section (p. 9, lines 169-172): 

“Given that the results of this study depend on the structure and physical properties of the test sample, the findings will only be applicable to the materials used in this study. Because these materials are standardized for porosity, density, purity, and mechanical strength as medical devices [12-14], the results will be reproducible within these limits.”

Comment 9: As a general comment, the work presented in this manuscript is very limited in its scope. For example, how does duration of impregnation affect physical, mechanical, and other properties of the CaP-based composite that are crucial for use in neurotrauma surgery (ATMOS group versus DECOMP group)? What is the impact on duration of the surgery?

This shortcoming of the work reported in this manuscript makes it very difficult to assess the extent to which the present findings will influence surgical practice (if at all).

Response: We appreciate the reviewer’s comment. Accordingly, we have added the following text to the Discussion section (p. 11, lines 217-222):

“In our neurotrauma surgery, antibiotic-impregnated calcium phosphate materials are prepared within approximately one hour from the onset to completion of the craniotomy. Accordingly, the completion of impregnation within 15 min, as shown in this study, would have no effect on neurotrauma surgery, and the sintered HAp material shows little dissolution to affect physical properties and mechanical strength in such a short duration of impregnation [19].”

Thank you again for your comments on our paper. We trust that the revised manuscript is suitable for publication.

RESPONSE TO REVIEWER 2:

We wish to express our appreciation to the Reviewer for his or her insightful comments, which have helped us significantly improve the paper.

Comment 1: Difference in impregnation ratio between hydroxyapatite and b-TCP with similar porosity should be discussed in more detail, especially D and E in Fig. 3. 

Response: We appreciate the reviewer’s comment. We agree that this point requires clarification, and have added the following text to the Discussion section (p. 10, lines 187-188):

“The difference in impregnation ratio between CP-D and CP-E despite their similar porosity may be due to differences in such microstructures within the materials.”

Thank you again for your comments on our paper. We trust that the revised manuscript is suitable for publication.

---

## [Decision Letter · Decision Letter 1]

3 Mar 2020

Atmospheric impregnation behavior of calcium phosphate materials for antibiotic therapy in neurotrauma surgery

PONE-D-19-30682R1

Dear Dr. KATO,

We are pleased to inform you that your manuscript has been judged scientifically suitable for publication and will be formally accepted for publication once it complies with all outstanding technical requirements.

With kind regards,

Esmaiel Jabbari, PhD

Academic Editor

PLOS ONE

Additional Editor Comments (optional):

Reviewers' comments:

Reviewer's Responses to Questions

**Comments to the Author**

1. If the authors have adequately addressed your comments raised in a previous round of review and you feel that this manuscript is now acceptable for publication, you may indicate that here to bypass the “Comments to the Author” section, enter your conflict of interest statement in the “Confidential to Editor” section, and submit your "Accept" recommendation.

Reviewer #1: (No Response)

Reviewer #2: All comments have been addressed

2. Is the manuscript technically sound, and do the data support the conclusions?

Reviewer #1: Yes

Reviewer #2: Yes

3. Has the statistical analysis been performed appropriately and rigorously? 

Reviewer #1: Yes

Reviewer #2: Yes

4. Have the authors made all data underlying the findings in their manuscript fully available?

Reviewer #1: Yes

Reviewer #2: Yes

5. Is the manuscript presented in an intelligible fashion and written in standard English?

Reviewer #1: Yes

Reviewer #2: Yes

6. Review Comments to the Author

Reviewer #1: The author's effort in including a brief review of the relevant literature is noted. However, the authors included comments on articles by only one research group (Itokazu et al. [Ref. #s 9-11]).

Critical comments on relevant article(s) from at least one other author should be included.

Reviewer #2: This paper is well revised according to the reviewers' comments. So it is acceptable for publication.

7. PLOS authors have the option to publish the peer review history of their article (what does this mean?). If published, this will include your full peer review and any attached files.

Reviewer #1: No

Reviewer #2: No

---

## [Editor Report · Acceptance letter]

5 Mar 2020

PONE-D-19-30682R1 

Atmospheric impregnation behavior of calcium phosphate materials for antibiotic therapy in neurotrauma surgery 

Dear Dr. KATO:

I am pleased to inform you that your manuscript has been deemed suitable for publication in PLOS ONE. Congratulations! Your manuscript is now with our production department. 

With kind regards,

on behalf of

Dr. Esmaiel Jabbari 

Academic Editor

PLOS ONE